# Shear Wave Splitting and Polarization in Anisotropic Fluid-Infiltrating Porous Media: A Numerical Study

**DOI:** 10.3390/ma13214988

**Published:** 2020-11-05

**Authors:** Nico De Marchi, WaiChing Sun, Valentina Salomoni

**Affiliations:** 1Department of Civil, Environmental and Architectural Engineering, University of Padua, Via Marzolo 9, 35131 Padua, Italy; nico.demarchi@dicea.unipd.it; 2Department of Civil Engineering and Engineering Mechanics, Columbia University, 614 SW Mudd, 4709, New York, NY 10027, USA; wsun@columbia.edu; 3Department of Management and Engineering, University of Padua, Stradella San Nicola 3, 36100 Vicenza, Italy

**Keywords:** porous media, Biot’s theory, acoustics, shear wave splitting, slow wave

## Abstract

The triggering and spreading of volumetric waves in soils, namely pressure (P) and shear (S) waves, developing from a point source of a dynamic load, are analyzed. Wave polarization and shear wave splitting are innovatively reproduced via a three-dimensional Finite Element research code upgraded to account for fast dynamic regimes in fully saturated porous media. The mathematical–numerical model adopts a u-v-p formulation enhanced by introducing Taylor–Hood mixed finite elements and the stability features of the solution are considered by analyzing different implemented time integration strategies. Particularly, the phenomena have been studied and reconstructed by numerically generating different types of medium anisotropy accounting for (i) an anisotropic solid skeleton, (ii) an anisotropic permeability tensor, and (iii) a Biot’s effective stress coefficient tensor. Additionally, deviatoric-volumetric coupling effects have been emphasized by specifically modifying the structural anisotropy. A series of analyses are conducted to validate the model and prove the effectiveness of the results, from the directionality of polarized vibrations, the anisotropy-induced splitting, up to the spreading of surface waves.

## 1. Introduction

The propagation of shear (S) and compressive/pressure (P) waves in poroelastic materials are phenomena relevant for seismology [1,2,3,4,5,6], geotechnical earthquake engineering [7,8,9,10,11], reservoir management [12], and biomechanics of bones and tissues [13]. The shear wave is a polarized transversal wave that propagates within the solid skeleton perpendicularly to the direction of the wave propagation and it often propagates slower than the pressure wave. When a shear wave bumps into an anisotropic medium, it may split into two or more waves with different speeds and orientations, a phenomenon known as shear wave splitting [14]. In particular, if we consider a transversely isotropic medium hit by a shear wave, the wave splits into two orthogonal polarized shear waves propagating at different velocities and orientations according to the material symmetry axis which may not coincide with the initial propagation direction.

The polarization of three-component shear wavetrains carry unique information about the internal structure of the rock: specifically, commonly observed shear-wave splitting may contain information about the orientation of crack distributions.

This information cannot usually be recovered from shear waves recorded at the free surface (i.e., Rayleigh ones) because of interference with the interaction of the shear wave with the surface, even for nearly vertical incidence. However, shear-wave splitting in synthetic three-component vertical seismic profiles in some cases may be interpreted directly in terms of the direction of the strike of vertical cracks [12]. The evolution of such fluid-saturated micro-cracks under changing conditions can be modeled by anisotropic poro-elasticity (APE). Numerical modeling with APE approximately matches a huge range of phenomena, including the evolution of shear-wave splitting during earthquake propagation, and enhanced oil recovery operations [15,16]. APE assumes, and recent observations of shear-wave splitting confirm, that the fluid-saturated cracks in the crust and (probably) upper mantle are so closely spaced that the cracked rocks are highly compliant critical systems with self-organized criticality [17].

Estimating the orientations of cracks, and hence of preferred directions of flow, by seismic investigations could be of crucial importance to production and reservoir engineering [12], as well as evaluating the influence of cracks on the effective elasticity of fractured rocks [18,19]. Additionally, polarization anomalies in seismic shear wavetrains, diagnostic of propagation through anisotropic media, have been observed in dilatancy zones in seismic regions. Stress-induced dilatancy will open cracks with preferred orientations, which will be effectively anisotropic to short-period seismic waves. The polarization anomalies are due to the shear waves splitting, in propagation through anisotropic media, into components with different polarizations and different velocities. This polarization writes characteristic signatures into the shear wavetrains [20].

Different numerical techniques can be used for analyzing wave propagation and shear wave splitting within a porous medium, such as finite difference/finite volume method [21], pseudospectral method [22], and finite element [23]. From the theoretical point of view, pioneer works by Biot are to be recalled [1], whereas in recent years the approaches by [5,10,24,25] can be of reference when analyzing localization and softening effects in wave propagation, or [26] for the contribution of anisotropy.

Here, the triggering and spreading of compressive waves in soils from a point source of a dynamic load are analyzed. Our focus is on the on polarization and shear wave splitting due to the anisotropy of the permeability tensor, the anisotropy of the solid skeleton, as well as to the novel Biot’s tensor that leads to an aniostropic effect on the hydro-mechanical coupling within the effective stress principle. The adopted multi-field dynamics poromechanics [27] Finite Element code is an upgraded dynamic version of a previous static one [28] able to describe the coupled hydro-mechanical behaviour of geomaterials. Particularly, two different solvers have been implemented to perform the time-space integration: the first considers Taylor–Hood elements together with an implicit Euler scheme, the second adopts equal-order elements and a semi-explicit-implicit scheme. A stabilization procedure based on the pressure projection method is used to circumvent the lack of inf-sup condition and resolve the sharp pore pressure gradient [29,30,31,32,33,34]. Comparisons are performed by taking into account literature examples and efficiency features are analyzed. The novel characteristics of the model lie in the synthesis of the compressibility of the fluid phase and the anisotropic permeability, as well as in the extension of the effective stress principle that predicts the anisotropy coupling of the partial stress of solid skeleton and that of pore fluid. One- and two-dimensional analyses have been conducted to validate the model against literature results, whereas a series of three-dimensional simulations are used to show the wave propagation of porous media that exhibits different types and combination of anisotropy.

As for notations and symbols, bold-faced letters denote tensors (including vectors which are rank-one tensors); the symbol ’·’ denotes a single contraction of adjacent indices of two tensors (e.g., $a\;\cdot \;b=a_{i}b_{i}$ or $c\;\cdot \;d=c_{ij}d_{jk}$ ); the symbol ‘:’ denotes a double contraction of adjacent indices of tensor of rank two or higher ( e.g., $c:\epsilon^{e}$ = $C_{ijkl}\epsilon_{kl}^{e}$ ); the symbol ‘⊗’ denotes a juxtaposition of two vectors (e.g., $a⊗b=a_{i}b_{j}$) or two symmetric second order tensors (e.g., ${\left(\alpha ⊗\beta \right)}_{ijkl}=\alpha_{ij}\beta_{kl}$). Moreover, ${\left(\alpha ⊕\beta \right)}_{ijkl}=\alpha_{jl}\beta_{ik}$ and ${\left(\alpha ⊖\beta \right)}_{ijkl}=\alpha_{il}\beta_{jk}$. We also define identity tensors ${\left(I\right)}_{ij}=\delta_{ij}$, ${\left(I^{4}\right)}_{ijkl}=\delta_{ik}\delta_{jl}$, and ${\left(I_{\mathrm{sym}}^{4}\right)}_{ijkl}=\frac{1}{2}\left(\delta_{ik}\delta_{jl}+\delta_{il}\delta_{kj}\right)$, where $\delta_{ij}$ is the Kronecker delta.

## 2. Mathematical Model

This section provides a brief review on the field equations for dynamic poromechanics problems, the finite element discretization, and the monolithic and semi-implicit scheme used to simulate the wave propagation in porous media (Section 2.1). This review is followed by the constitutive laws that replicate the anisotropy of the solid skeleton elasticity, and those of the hydraulic responses (permeability), of hydromechanical coupling mechanisms (tensorial Biot’s coefficient).

### 2.1. Governing Equations

The porous media are here modeled following the mixture theory where the existence of an effective medium of a size suitable for the Representative Elementary Volume (REV) is assumed. A fully saturated porous material is then represented by a corresponding effective medium in which the specific distributions of the fluid and solid constituents inside the REV are homogenized such that the volume of each material point is occupied by a fraction of solid and fluid constituents (Figure 1). The balance of mass and linear momentum equations in the dynamic regime read
(1)$$\begin{array}{c} {\dot{\left(\rho^{\alpha }\right)}}_{\alpha }+\rho^{\alpha }\nabla^{x}\cdot v_{\alpha }=0\;; \end{array}$$
(2)$$\begin{array}{c} \rho^{\alpha }{\dot{\left(v_{\alpha }\right)}}_{\alpha }=\nabla^{x}\cdot \sigma^{\alpha }+\rho^{\alpha }g+h^{\alpha }\;, \end{array}$$
where: $\rho^{\alpha }=n^{\alpha }\rho_{\alpha }$ is the partial mass density of α-phase (i.e., solid S or fluid F) computed as a product of the volume fraction $n_{\alpha }$ and the intrinsic mass density of α constituent (The *Stanford notation* is adopted here to indicate the partial quantities (superscript index) and intrinsic quantities (subscript index))—with: $\sum_{\alpha }\rho^{\alpha }=\rho$. $v_{\alpha }$ being the velocity vector, $\sigma^{\alpha }$ the partial Cauchy stress tensor of α phase, g is the gravitational acceleration, and $h^{\alpha }$ is the volume specific local interaction force between the phases, so that: $\sum_{\alpha }h^{\alpha }=0$. Furthermore, the spatial gradient operator error and the material time derivative ${\dot{\left(*\right)}}_{\alpha }$ are referred to as: (3)$$\begin{array}{c} \nabla^{x}\left(*\right)=\frac{\partial (*)}{\partial x}\;;\; \end{array}$$(4)$$\begin{array}{c} {\dot{\left(*\right)}}_{\alpha }=\frac{d_{\alpha }\left(*\right)}{dt}=\frac{\partial (*)}{\partial t}+\nabla^{x}\left(*\right)\;\cdot \;v_{\alpha }\;, \end{array}$$

No thermal effects nor mass exchanges between phases are taken into account; the bulk modulus of the α-phase can be defined as:(5)$$K_{\alpha }=\rho_{\alpha }\frac{dp_{\alpha }}{d\rho_{\alpha }}\;$$
where $p_{\alpha }$ is the intrinsic Cauchy pressure of the α-phase.

### 2.2. Constitutive Laws

#### 2.2.1. Effective Stress Law for Anisotropic Elasticity

It is assumed to consider a homogeneous, linear elastic, anisotropic porous medium so that the stress–strain law can be written as:(6)σ=C:ϵ
where c is the fourth-order elastic constitutive tensor of the mixture and its components depend on the material symmetries; when adopting Voigt notation:(7)$$\bar{\sigma }={\left[\begin{array}{cccccc} \sigma_{xx} & \sigma_{yy} & \sigma_{zz} & \sigma_{xy} & \sigma_{yz} & \sigma_{zx} \end{array}\right]}^{T};\;\bar{\epsilon }={\left[\begin{array}{cccccc} \epsilon_{xx} & \epsilon_{yy} & \epsilon_{zz} & \epsilon_{xy} & \epsilon_{yz} & \epsilon_{zx} \end{array}\right]}^{T};$$
a transversely isotropic material is characterized by the 6 × 6 compliance matrix:(8)$$\bar{S}=\left[\begin{array}{cccccc} 1/E_{h} & -\nu_{hh}/E_{h} & -\nu_{vh}/E_{v} & 0 & 0 & 0\\ -\nu_{hh}/E_{h} & 1/E_{h} & -\nu_{vh}/E_{v} & 0 & 0 & 0\\ -\nu_{hv}/E_{h} & -\nu_{hv}/E_{h} & 1/E_{v} & 0 & 0 & 0\\ 0 & 0 & 0 & 1/G_{hh} & 0 & 0\\ 0 & 0 & 0 & 0 & 1/G_{hv} & 0\\ 0 & 0 & 0 & 0 & 0 & 1/G_{vh} \end{array}\right]$$

The model exhibits symmetry along any direction *h* belonging to the isotropy plane and different material properties along the normal direction *v*. The independent constants are: $E_{h},E_{v},G_{hv},\nu_{hh},\nu_{hv}$, being the other terms given by:(9)$$\nu_{hv}/E_{h}=\nu_{vh}/E_{v}\;;\;G_{hh}=E_{hh}/\left(2\left(1+\nu_{hh}\right)\right).$$

Generally, the symmetry axes for an anisotropic material with respect to the reference system are not coaxial with the global axes so that the elastic matrix has all non-zero elements; hence, to express stress and strain measures into the reference system, the 6×6 transformation matrix is adopted: (10)$$\bar{M}=\left[\begin{array}{cccccc} r_{11}^{2} & r_{21}^{2} & r_{31}^{2} & 2r_{11}r_{21} & 2r_{21}r_{31} & 2r_{11}r_{31}\\ r_{12}^{2} & r_{22}^{2} & r_{32}^{2} & 2r_{12}r_{22} & 2r_{22}r_{32} & 2r_{12}r_{32}\\ r_{13}^{2} & r_{23}^{2} & r_{33}^{2} & 2r_{13}r_{23} & 2r_{23}r_{33} & 2r_{13}r_{33}\\ r_{11}r_{12}\; & r_{21}r_{22}\; & r_{31}r_{32}\; & r_{11}r_{22}+r_{21}r_{12}\; & r_{21}r_{32}+r_{31}r_{22}\; & r_{11}r_{32}+r_{31}r_{12}\\ r_{12}r_{13}\; & r_{22}r_{23}\; & r_{32}r_{33}\; & r_{12}r_{23}+r_{22}r_{13}\; & r_{22}r_{33}+r_{32}r_{23}\; & r_{12}r_{33}+r_{32}r_{13}\\ r_{11}r_{13}\; & r_{21}r_{23}\; & r_{31}r_{33}\; & r_{11}r_{23}+r_{21}r_{13}\; & r_{21}r_{33}+r_{31}r_{23}\; & r_{11}r_{33}+r_{31}r_{13} \end{array}\right];\;R=\left[\begin{array}{ccc} r_{11} & r_{12} & r_{13}\\ r_{21} & r_{22} & r_{23}\\ r_{31} & r_{32} & r_{33} \end{array}\right]$$
so that:(11)$${\bar{C}}_{glob}=\bar{M}{\bar{C}}_{loc}{\bar{M}}^{T},\;or\;{\bar{S}}_{glob}={\bar{M}}^{-T}{\bar{S}}_{loc}{\bar{M}}^{-1}\;;$$
with the bar above the symbols indicating the matrix version of the tensor quantities.

#### 2.2.2. Tensorial Nature of Biot’s Coefficient

Here, the effective stress principle is adopted, where the Biot’s effective stress coefficient is not a scalar but a second-order tensor with different eigenvalues (see Cowin and Doty [26], Carroll [35]):(12)$$\sigma_{E}^{S}=\sigma +Ap_{F}\;;\;A=\left(I-C\;S_{S}\right):1,$$
where $\sigma_{E}^{S}$ is the effective stress tensor, $\sigma =\sigma^{S}+\sigma^{F}$ is the total stress tensor sum of the two partial stresses, $p_{F}$ is the Cauchy pore pressure, 1 and I are the unit and the identity tensors, respectively, whereas $S_{s}$ refers to the solid skeleton. Equation (Equation 12) represents an extension of the classical effective stress concept, based on the assumption of isotropy for both porous material and solid phase. The expression comes from analyzing the constitutive stress–strain behaviour of the REV by assuming structural (anisotropic porous geometry), intrinsic (anisotropic solid material) or both anisotropies. As described in Cowin and Doty [26], the Biot’s effective stress tensor could have six nonzero components if the material symmetry of compliance tensor $S_{S}$ is less than transversely isotropic and/or its axis of symmetry are not coaxial with axis of the transversely isotropic model of the porous material C. If both porous material and its solid skeleton are isotropic, we obtain that:(13)$$A=B1\;;\;B=\left[1-(K/K_{S})\right]\;$$
where *B* is the classic Biot’s effective stress coefficient [36], equal to one if incompressibility of the solid skeleton is assumed.

#### 2.2.3. Darcy’s Law

By assuming laminar flow, the generalized form of Darcy’s law is considered, relating the fluid mass flow rate to the pore pressure gradient:(14)$$n^{F}w_{F}=-K^{F}\left(\nabla^{x}p_{F}-\rho_{F}g\right)\;$$
where $w_{F}=v_{F}-w_{S}$ is the relative velocity and $w^{F}=n^{F}w_{F}$ the Darcy velocity. The quantity $K^{F}$ is the second-order specific permeability tensor:(15)$$K^{F}=k^{F}/\gamma_{F}={\hat{K}}^{F}/\mu_{F}\;$$
where $\gamma_{F}=\rho_{F}g$ is the fluid intrinsic unit weight, $\mu_{F}$ is the fluid viscosity, $k^{F}$ is the hydraulic conductivity tensor, and ${\hat{K}}^{F}$ is the intrinsic permeability. The latter is a second-order tensor of which the eigenvalues and spectral directions can be determined from experimental filtration tests or through 3D tomography images and geometrical analyses as used in Sun et al. [37].

## 3. Numerical Implementation

### 3.1. Galerkin Form

The subsequent set of coupled partial differential equations (PDEs) is hence obtained: (16)$$\begin{array}{c} {\dot{\left(u_{S}\right)}}_{S}=v_{S}\;; \end{array}$$(17)$$\begin{array}{c} \rho^{S}{\dot{\left(v_{S}\right)}}_{S}=\nabla^{x}\left[\sigma_{E}^{S}-\left(A-n^{F}1\right)p_{F}\right]+\rho^{S}g+\frac{{\left(n^{F}\right)}^{2}\gamma_{F}}{K^{F}}w_{F}-p_{F}\nabla^{x}\left(n^{F}\right)\;; \end{array}$$(18)$$\begin{array}{c} \rho^{F}{\dot{\left(v_{F}\right)}}_{S}+\rho_{F}\nabla^{x}\left(v_{F}\right)w_{F}=-n^{F}\nabla^{x}\left(p_{F}\right)+\rho^{F}g-\frac{{\left(n^{F}\right)}^{2}\gamma_{F}}{K^{F}}w_{F}\;; \end{array}$$(19)$$\begin{array}{c} \frac{n^{S}}{K_{S}}{\dot{\left(p_{S}\right)}}_{S}+\nabla^{x}v_{S}+\frac{n^{F}}{K_{F}}{\dot{\left(p_{F}\right)}}_{S}+\frac{1}{\rho_{F}}\nabla^{x}\left(\rho^{F}w_{F}\right)=0\;. \end{array}$$
where the unknowns of the coupled problem are $u_{S}$, $v_{S}$, $v_{F}$, and $p_{F}$. This set of equations represents the three-field formulation (it is recalled that $v_{S}$ are secondary unknowns and the first equation is written to compute them). Furthermore, if we consider that the porous material is subjected to low and relatively small frequencies (≤30 Hz in the geomechanical problems), we can assume that ${\dot{\left(u_{S}\right)}}_{S}\approx {\dot{\left(u_{F}\right)}}_{S}$, and ${\dot{\left(w_{F}\right)}}_{S}\approx 0$ so we can rearrange the equations and obtain the classical two-field formulation for consolidation problems with $u_{S}$ and $p_{F}$. The governing equations become:(20)$$\begin{array}{c} {\dot{\left(u_{S}\right)}}_{S}=v_{S}\;; \end{array}$$(21)$$\begin{array}{c} \rho {\dot{\left(v_{S}\right)}}_{S}=\nabla^{x}\left(\sigma_{E}^{S}-Ap_{F}\right)+\rho g; \end{array}$$(22)$$\begin{array}{c} \frac{n^{S}}{K_{S}}{\dot{\left(p_{S}\right)}}_{S}+\nabla^{x}v_{S}+\frac{n^{F}}{K_{F}}{\dot{\left(p_{F}\right)}}_{S}+\frac{1}{\rho_{F}}\nabla^{x}\left(\rho^{F}w_{F}\right)=0\;. \end{array}$$

By considering a finite domain Ω of the mixture with its boundary Γ and a set of independent test functions, the weak formulation for the previous PDEs can be written as: (23)$$\begin{array}{cc} G_{v_{S}}\left(u_{S},v_{S},\delta u_{S}\right)= & \int_{\Omega }\delta u_{S}\;\cdot \;\left[{\dot{\left(u_{S}\right)}}_{S}-v_{S}\right]dv=0\;;\\ G_{u_{S}}\left(u_{S},v_{S},v_{F},p_{F},\delta u_{S}\right)= & \int_{\Omega }\nabla^{x}\left(\delta u_{S}\right)\;\cdot \;\left[\sigma_{E}^{S}-\left(A-n^{F}I\right)p_{F}\right]dv+\int_{\Omega }\delta u_{S}\;\cdot \;\left[-\frac{{\left(n^{F}\right)}^{2}\gamma_{F}}{K^{F}}w_{F}+p_{F}\nabla^{x}\left(n^{F}\right)\right]dv \end{array}$$(24)$$\begin{array}{cc} \; & +\int_{\Omega }\delta u_{S}\;\cdot \;\left\{\rho^{S}\left[{\dot{\left(v_{S}\right)}}_{S}-g\right]\right\}dv-\int_{\partial \Omega_{S}}\delta u_{S}\;\cdot \;t^{S}da=0\;;\\ G_{v_{F}}\left(u_{S},v_{S},v_{F},p_{F},\delta v_{F}\right)= & -\int_{\Omega }\nabla^{x}\left(\delta v_{F}\right)n^{F}p_{F}dv-\int_{\partial \Omega_{F}}\delta v_{F}\;\cdot \;t^{F}da \end{array}$$
(25)$$\begin{array}{c} +\int_{\Omega }\delta v_{F}\;\cdot \;\left\{\rho_{F}\left[{\dot{\left(v_{F}\right)}}_{S}-g+\left(\nabla^{x}v_{F}+\frac{n^{F}\gamma_{F}}{K^{F}}\right)w_{F}\right]\right\}dv=0\;; \end{array}$$
(26)$$\begin{array}{cc} H_{p_{F}}\left(v_{S},v_{F},p_{F},\delta p\right)= & \int_{\Omega }\delta pA\;:\;\nabla^{x}v_{S}dv+\int_{\Omega }\delta p\Lambda {\dot{\left(p_{F}\right)}}_{S}dv-\int_{\Omega }\nabla^{x}\left(\delta p\right)\;\cdot \;\left(n^{F}w_{F}\right)dv+\int_{\partial \Omega_{p}}\delta p\;\cdot \;\bar{v}da=0\;. \end{array}$$
where $\delta u_{S}$, $\delta u_{F}$, $\delta v_{F}$, and δp are the test functions used as weight. The quantities $t^{S}$ and $t^{F}$ are the surface tractions of the single phases acting on the Neumann boundary and $\bar{v}$ is the volume flux of the fluid going through the boundary of the mixture. The compressibility modulus Λ is computed as:(27)$$\Lambda =\frac{n^{S}}{K_{S}}+\frac{n^{F}}{K_{F}}-A:C:A=\frac{n^{S}}{K_{S}}+\frac{n^{F}}{K_{F}}-1:C_{S}Sc_{S}:1;$$
this expression is the same as Equation (8.5) in Cowin [38].

In order to write the Galerkin formulation in a compact way, Equations (Equation 23)–(Equation 26) and all the field variables are collected within vectors:(28)$$G_{u}=\left[\begin{array}{c} G_{v_{S}}\\ G_{u_{S}}\\ G_{v_{F}}\\ G_{p_{F}} \end{array}\right],\;u=\left[\begin{array}{c} u_{S}\\ v_{S}\\ v_{F}\\ p_{F} \end{array}\right],\;\delta u=\left[\begin{array}{c} \delta u_{S}\\ \delta u_{S}\\ \delta v_{F}\\ \delta p_{F} \end{array}\right],\;{\dot{\left(u\right)}}_{S}=\left[\begin{array}{c} {\dot{\left(u_{S}\right)}}_{S}\\ {\dot{\left(v_{S}\right)}}_{S}\\ {\dot{\left(v_{F}\right)}}_{S}\\ {\dot{\left(p_{F}\right)}}_{S} \end{array}\right],\;u_{0}=\left[\begin{array}{c} u_{S0}\\ v_{S0}\\ v_{F0}\\ p_{F0} \end{array}\right].$$

The spatial discretization is carried out via the Finite Element Method, referring to the following discrete unknown variables and test functions, respectively:(29)$$u^{h}\left(x,t\right)={\bar{u}}^{h}+\sum_{i=1}^{N_{u}}N_{ui}\left(x\right)u_{i}\left(t\right),\;\delta u^{h}\left(x\right)=\sum_{i=1}^{M_{u}}M_{ui}\left(x\right)\delta u_{i}\left(t\right);$$
where ${\bar{u}}^{h}$ represents the solution on the Dirichlet boundary, $N_{u}$ and $M_{u}$ are the numbers of nodes, and $N_{ui}$ and $M_{ui}$ indicate the shape functions at node *i* depending only on the position x. The final three-field variational problem can be re-written via the Petrov–Galerkin formulation $G_{u}\left(\delta u^{h},u^{h}\right)=0$ using the previous test and trial functions.

### 3.2. Matrix Form and Time Discretization

By omitting the superscript *h* for vectors u and δu and indicating with $f_{S}$, $f_{F}$ and $f_{p}$ the space-discrete Neumann boundary terms, the subsequent matrix formulation can be obtained:(30)$$G_{u}^{h}=\left[\begin{array}{cccc} I & 0 & 0 & 0\\ 0 & M_{22} & 0 & 0\\ 0 & 0 & M_{33} & 0\\ 0 & 0 & 0 & M_{44} \end{array}\right]\left[\begin{array}{c} {\dot{u}}_{S}\\ {\dot{v}}_{S}\\ {\dot{v}}_{F}\\ {\dot{p}}_{F} \end{array}\right]+\left[\begin{array}{cccc} 0 & -I & 0 & 0\\ K_{21} & K_{22} & K_{23} & K_{24}\\ 0 & K_{32} & K_{33} & K_{34}\\ 0 & K_{42} & K_{43} & 0 \end{array}\right]\left[\begin{array}{c} u_{S}\\ v_{S}\\ v_{F}\\ p_{F} \end{array}\right]-\left[\begin{array}{c} 0\\ a_{S}+f_{S}\\ a_{F}+f_{F}\\ f_{p} \end{array}\right]=0\;$$

The system (Equation 30) is composed by a set of pure differential equations where the time evolution for all the primary variables can be written as a first-order ordinary differential equation:(31)$$G_{u}^{h}=M\dot{u}+Ky-f=0;\;\dot{u}=g\left(t,u\right),\;and\;y\left(t_{0}\right)=y_{0}$$

This formulation can be solved via implicit time integration schemes or explicit ones. In the latter case, the stability constraint must be taken into consideration when choosing the time step size. In the case where both solid and fluid constituents are incompressible, the system (Equation 30) becomes a differential algebraic one, where the fourth relation turns into a volume balance equation for the mixture acting as an algebraic incompressibility constraint. In order to solve a saddle point problem, the time integration schemes must be upgraded by special numerical techniques to maintain the stability of the numerical solution. Following Markert et al. [39], two numerical time integration strategies have been adopted: the first one is an implicit monolithic scheme together with mixed order interpolation for the primary variables; the second one is a semi-explicit/implicit splitting scheme together with an equal order interpolation for the unknowns.

#### 3.2.1. Implicit Monolithic Schemes

Different types of implicit schemes exist and it is possible to subdivide them mainly into two classes: one step and multi-step methods. Thanks to the lower computational cost and memory storage of the variables, only the first class is considered here. In Algorithm 1, the pseudo code of the generalized trapezoidal scheme inside a Newton–Raphson procedure is shown. By considering the integration variable θ=1, the Backward Euler scheme is obtained; this method is first order accurate but small time steps are mandatory in order to reduce the artificial numerical damping leading to wrong solution (see Jansen et al. [40]). For θ=1/2, the second-order accurate Crank–Nicholson scheme comes out, and with θ=0 the Explicit Forward Euler scheme appears.

Note that the last equation of the coupled system (Equation 30) is modified by inserting the expression of Darcy’s velocity derived from the fluid momentum balance in order to improve the numerical stability of the scheme.

**Algorithm 1:** Newton–Raphson Algorithm.Initialization: $\;u_{n}={\left[u_{Sn}\;v_{Sn}\;v_{Fn}\;p_{n}\right]}^{T}=0,\;F_{n}={\left[0\;f_{Sn}\;f_{Fn}\;q_{n}\right]}^{T}=0$for $n=1:n_{end}$
 $\;F_{n+1}=F_{n}+\theta \Delta F,\;u_{n+1}=u_{n},$
  for $i=1:i_{max}$

 $\;\Delta u_{n+1}^{i}=u_{n+1}^{i}-u_{n}$
   Compute matrix:$\;M_{ii},\;K_{ii}$   Compute residual:
$R^{i+1}=\left[\begin{array}{c} 0\\ r_{v_{S}}\\ r_{v_{F}}\\ r_{p_{F}} \end{array}\right]=F_{n+1}+\left[\begin{array}{c} 0\\ a_{2}\\ a_{3}\\ a_{4} \end{array}\right]-\left[\begin{array}{cccc} I & 0 & 0 & 0\\ 0 & M_{22} & 0 & 0\\ 0 & 0 & M_{33} & 0\\ 0 & 0 & M_{43} & M_{44} \end{array}\right]\;\frac{\Delta u_{n+1}^{i}}{\Delta t}-\left[\begin{array}{cccc} 0 & -I & 0 & 0\\ K_{21} & K_{22} & K_{23} & K_{24}\\ 0 & K_{32} & K_{33} & K_{34}\\ 0 & K_{42} & K_{43} & K_{44} \end{array}\right]\;\left[\begin{array}{c} \theta u_{n+1}^{i}+\left(1-\theta \right)u_{n} \end{array}\right]$
   Check residual:    if$\;R^{i+1}<toll$    break    end   Compute Jacobi matrix:
 J=M/Δt+θK
   Compute Y-increment:
 $\;du=J^{-1}R^{i+1}$
   Update solution:
 $\;u_{n+1}^{i+1}=u_{n+1}^{i}+du$
  endend

#### 3.2.2. Semi-Explicit/Implicit Splitting Scheme

This scheme solves the coupled problem using a splitting procedure applied in the field of porous mechanics by Huang et al. [41,42]. Through this scheme, the system of equations is split into an implicit and a subsequent explicit step. This method is restricted by a critical time-step to guarantee stability and accuracy of the solution. The splitting separates the linear momentum balances from the mass balance of the mixture and decouples the displacement and velocity fields from the pore-fluid pressure field. To this aim, the time discretization of the governing equations is to be developed before the spatial one together with the definition of an intermediate velocity giving an approximation of the velocities of the phases in the next time step. By following the same procedure of Markert et al. [39], the time discretization and splitting procedure starts with the implicit discretization of the solid velocity. Equation (Equation 16) becomes through the trapezoidal rule:(32)$$\frac{\left(u_{Sn+1}-u_{Sn}\right)}{\Delta t_{n}}=\frac{1}{2}\left(v_{Sn+1}+v_{Sn}\right)\;;$$
assuming that $n^{\alpha }\approx n_{0}^{\alpha }$, $\nabla^{x}n^{\alpha }\approx 0$ (small strain assumption); by explicitly considering the solid extra stress tensor $\sigma_{En}^{S}=\sigma_{E}^{S}\left(u_{Sn}\right)$ and implicitly the pore-fluid pressure term $p_{F}$ together with the relative velocities through the intermediate velocities $v_{S}^{*}$ and $v_{F}^{*}$, the momentum balance equation for solid phase is rearranged and split as follows: (33)$$\begin{array}{c} \rho^{S}\frac{\left(v_{S}^{*}-v_{Sn}\right)}{\Delta t_{n}}=\nabla^{x}\sigma_{En}^{S}-\left(A-n^{F}1\right)\nabla^{x}\left(p_{Fn}\right)+\rho^{S}g+\frac{{\left(n^{F}\right)}^{2}\gamma_{F}}{K^{F}}w_{F}^{*}\;; \end{array}$$(34)$$\begin{array}{c} \rho^{S}\frac{\left(v_{Sn+1}-v_{S}^{*}\right)}{\Delta t_{n}}=-\left(A-n^{F}1\right)\nabla^{x}\left(p_{Fn+1}-p_{Fn}\right)\;. \end{array}$$

The momentum balance for the fluid becomes: (35)$$\begin{array}{c} \rho^{F}\frac{\left(v_{F}^{*}-v_{Fn}\right)}{\Delta t_{n}}+\rho^{F}\left(\nabla^{x}v_{F}^{*}\right)w_{F}^{*}=-n^{F}\nabla^{x}\left(p_{Fn}\right)+\rho^{F}g-\frac{{\left(n^{F}\right)}^{2}\gamma_{F}}{K^{F}}w_{F}^{*}\;; \end{array}$$(36)$$\begin{array}{c} \rho^{F}\frac{\left(v_{Fn+1}-v_{F}^{*}\right)}{\Delta t_{n}}=-n^{F}\nabla^{x}\left(p_{Fn+1}-p_{Fn}\right)\;; \end{array}$$
where $\rho^{F}\left(\nabla^{x}v_{F}^{*}\right)w_{F}^{*}$ is a convective term. The mass balance is expressed as:(37)$$\Lambda \frac{p_{Fn+1}-p_{Fn}}{\Delta t}+A:\nabla^{x}\left(v_{Sn+1}\right)+\frac{1}{\rho_{F}}\nabla^{x}\left(\rho^{F}w_{Fn+1}\right)=0\;.$$

Rebuilding the weak formulation, for the solid phase: $$\begin{array}{cc} \int_{\Omega }\delta u_{S}\;\cdot \;\left[\frac{\left(v_{S}^{*}-v_{Sn}\right)}{\Delta t_{n}}-g\right]n^{S}\rho_{S}dv-\int_{\Omega }\delta u_{S}\;\cdot \;\frac{{\left(n^{F}\right)}^{2}\gamma_{F}}{K^{F}}w_{F}^{*}dv & \end{array}$$(38)$$\begin{array}{cc} +\int_{\Omega }\nabla^{x}\left(\delta u_{S}\right)\;\cdot \;\left[\sigma_{En}^{S}-\left(A-n^{F}1\right)p_{Fn}\right]dv-\int_{\partial \Omega_{S}}\delta u_{S}\;\cdot \;t_{n}^{S}da=0 & \;; \end{array}$$(39)$$\begin{array}{cc} \int_{\Omega }\delta u_{S}\;\cdot \;\left[\rho^{S}\frac{\left(v_{Sn+1}-v_{S}^{*}\right)}{\Delta t_{n}}+\left(A-n^{F}1\right)\nabla^{x}\left(p_{Fn+1}-p_{Fn}\right)\right]dv=0 & \;; \end{array}$$
for the fluid phase: (40)$$\begin{array}{cc} \int_{\Omega }\delta v_{F}\;\cdot \;\left[\frac{\left(v_{F}^{*}-v_{Fn}\right)}{\Delta t_{n}}-g\right]n^{F}\rho_{F}dv+\int_{\Omega }\delta v_{F}\;\cdot \;n^{F}\nabla^{x}\left(p_{Fn}\right)dv+\int_{\Omega }\delta v_{F}\;\cdot \;\frac{n^{F}\gamma_{F}}{K^{F}}w_{F}^{*}dv=0 & \;; \end{array}$$(41)$$\begin{array}{cc} \int_{\Omega }\delta v_{F}\;\cdot \;\frac{\left(v_{Fn+1}-v_{F}^{*}\right)}{\Delta t_{n}}n^{F}\rho_{F}dv-\int_{\Omega }\nabla^{x}\left(\delta v_{F}\right)n^{F}\left(p_{Fn+1}-p_{Fn}\right)dv-\int_{\partial \Omega_{F}}\delta v_{F}\;\cdot \;t_{n+1}^{F}da=0 & \;; \end{array}$$
with the mass balance:(42)$$\int_{\Omega }\delta pA\;:\;\nabla^{x}v_{S}^{*}dv+\int_{\Omega }\delta p\Lambda \frac{\left(p_{Fn+1}-p_{Fn}\right)}{\Delta t_{n}}dv-\int_{\Omega }\nabla^{x}\left(\delta p\right)\;\cdot \;\left(n^{F}w_{F}\right)dv+\int_{\partial \Omega_{p}}\delta p\;\cdot \;{\bar{v}}_{n+1}da=0\;.$$

From the weak formulation the corresponding matrix equations are built following the scheme of Algorithm 2 and the expressions of all the matrices are plotted in Appendix A. After initialization, the procedure starts by explicitly computing the intermediate velocities $v^{*}$; then, the pore-fluid pressure is calculated and subsequently the velocity corrections and finally the solid displacements $u_{Sn+1}$. Clearly, each time-step requires an iteration check to be satisfied: $∥u_{n+1}^{i+1}-u_{n+1}^{i}∥<toll$, with the critical time-step given by (for a 3D linear FE):(43)$$\Delta t_{cr}=\frac{\Delta h_{x}\Delta h_{y}\Delta h_{z}}{c_{p}^{x}\Delta h_{x}+c_{p}^{y}\Delta h_{y}+c_{p}^{z}\Delta h_{z}}\;\;c_{p}^{r}=\sqrt{\frac{C_{rr}}{\rho^{S}}},\;r=x,y,z,$$
with $C_{rr}$ elastic matrix component of the porous material.


**Algorithm 2:** Prediction/Correction Algorithm.Initialization: $\;u_{Sn}=0,\;v_{n}=0,\;p_{n}=0$for $n=1:n_{end}$
 $\;f_{n}=f_{n}+\Delta f,\;v^{P}=v_{n},\;p^{P}=p_{n},$

 $\;\Delta u_{S}=\Delta tv_{S}^{P},\;u_{Sn}+\Delta u_{S}$
  for $i=1:i_{max}$   Compute matrix:$\;M,\;K_{ii},\;{\bar{K}}_{ii}$   Compute prediction velocities:
$\;\left[\begin{array}{c} v_{S}^{*}\\ v_{F}^{*} \end{array}\right]={\left[\begin{array}{cc} \frac{M_{22}}{\Delta t}+K_{22} & -K_{23}\\ -K_{32} & \frac{M_{33}}{\Delta t}+K_{33} \end{array}\right]}^{-1}\;\left\{\begin{array}{c} \left[\begin{array}{cc} \frac{M_{22}}{\Delta t} & 0\\ 0 & \frac{M_{33}}{\Delta t} \end{array}\right]\left[\begin{array}{c} v_{Sn}^{P}\\ v_{Fn}^{P} \end{array}\right]+\left[\begin{array}{cc} -K_{21} & K_{24}\\ 0 & -{\bar{K}}_{34} \end{array}\right]\left[\begin{array}{c} u_{Sn}^{P}\\ p_{n}^{P} \end{array}\right]+\left[\begin{array}{c} a_{2}\\ a_{3} \end{array}\right]+\left[\begin{array}{c} f_{Sn}\\ 0 \end{array}\right] \end{array}\right\}$
   Compute pore fluid pressure:
$\;\left[\begin{array}{c} 0\\ p_{n+1}^{*} \end{array}\right]=\left[\begin{array}{c} 0\\ p_{n}^{P} \end{array}\right]+{\left[\begin{array}{cc} I & 0\\ 0 & {\bar{K}}_{44} \end{array}\right]}^{-1}\;\left\{\begin{array}{c} \left[\begin{array}{cc} 0 & 0\\ -K_{42} & K_{43} \end{array}\right]\left[\begin{array}{c} v_{S}^{*}\\ v_{F}^{*} \end{array}\right]-\left[\begin{array}{c} 0\\ {\bar{f}}_{Pn+1} \end{array}\right] \end{array}\right\}$
   Compute velocities correction:
$\;\left[\begin{array}{c} v_{Sn+1}\\ v_{Fn+1} \end{array}\right]=\left[\begin{array}{c} v_{S}^{*}\\ v_{F}^{*} \end{array}\right]+{\left[\begin{array}{cc} \frac{M_{22}}{\Delta t} & 0\\ 0 & \frac{M_{33}}{\Delta t} \end{array}\right]}^{-1}\;\left\{\begin{array}{c} \left[\begin{array}{cc} 0 & {\bar{K}}_{24}\\ 0 & -K_{34} \end{array}\right]\left[\begin{array}{c} 0\\ p_{n+1}-p_{n}^{P} \end{array}\right]+\left[\begin{array}{c} 0\\ f_{Pn+1} \end{array}\right] \end{array}\right\}$
   Compute solid displacements:
$\;u_{Sn+1}=u_{Sn}+\frac{1}{2}\Delta t\left(v_{Sn+1}+v_{Sn}\right)$

$\;r_{i}=u_{Sn+1}-u^{P}$
    if$\;r_{i}<toll$    break    end   Update prediction variables:
$\;v^{P}=v_{n+1},\;u_{S}^{P}=u_{Sn+1},\;\Delta u_{S}=u_{Sn+1}-u_{Sn},$
  end end


## 4. Numerical Examples

In this section, some numerical analyses are described, accounting for different types of anisotropy affecting P- and S-wave motion within porous media.

Particularly, waves propagation and interaction are studied, influenced by (1) the induced anisotropy related to volumetric-deviatoric coupling effects and (2) the inherent anisotropy of transversely isotropic elasticity for the solid skeleton, the anisotropy of the permeability tensor, and the anisotropic hydro-mechanical coupling effect captured by the tensorial Biot’s approach. The numerical analyses have been developed via GeoMatFEM [28], a MATLAB research software suitable for coupled geo-mechanical simulations and now upgraded to a dynamic version. Two benchmarks are included to validate the implementation.

### 4.1. Benchmark Cases with Isotropic Elastic Materials

The code has been validated against two numerical examples:fully saturated soil column under harmonic load (cf. de Boer et al. [43]);wave propagation within a two-dimensional soil domain (cf. Markert et al. [39]).

The soil column model shown in Figure 2a is considered, with a vertical discretization of 10 Finite Elements/meter subjected to a vertical harmonic load Figure 2b. The material parameters are shown in Table 1.

The results obtained by adopting different solvers together with the analytical solution are depicted in Figure 3. The Implicit Backward Euler scheme B.E. (coupled with mixed elements), the semi-explicit/implicit scheme S.E. (with linear equal order elements), both considering the *uvp–formulation* and the classical *up–formulation*), have been taken into account. For the two implicit schemes, a time step $dt\;=\;0.5\times 10^{-3}\;s$ has been adopted, while, for the semi-explicit/implicit scheme, a $dt\;=\;2.5\times 10^{-4}\;s$ has been assumed. All of the numerical solutions give equal results in terms of compaction (see Figure 3a) and pore pressure (see Figure 3c); slight differences are observable in the peak values of the fluid velocity on surface (see Figure 3b), but this is due to the choice of the exit error tolerance of the schemes, while, for the effective stress (see Figure 3d), the differences come from the fact that they are computed at Gauss points 0.0225 m away from the reference surface.

As regards Benchmark (2), the F.E. domain together with the boundary conditions are shown in Figure 4a. The material parameters are the same as in the previous example (see Table 1) and the soil is subjected to an impulsive load Figure 4b, with H(t−τ) Heaviside function and τ=0.04s the duration of the impulse.

Figure 5 reports the main results by varying the numerical solver, i.e., the implicit Backward Euler scheme with mixed elements, considering both *uv* and *uvp–formulation*, the Semi-explicit/Implicit scheme with linear (L) and quadratic (Q) equal order elements. A composite time integration scheme between the trapezoidal rule and 2nd–order backward difference scheme (TR-BDF2) has been additionally adopted: this scheme is inserted in the Runge–Kutta method (s-stage DIRK) together with mixed elements. In Markert et al. [39], this scheme was applied for the first time in the field of multiphase porous mechanics, the propriety of such a method is that it satisfies all the stability requirements and it is second order accurate: for these reasons, its results have been taken as a reference solution. The same mesh discretization composed by 4 hexahedral elements per square meter is assumed for all the models; furthermore, for the three implicit schemes and for the semi-explicit one with linear elements, the time step dt = 10^−3^ s has been adopted, while for the semi-explicit scheme with quadratic elements, dt = 2.5 10^−4^ s has been assumed in order to satisfy the CFL stability condition.

Figure 5a shows the vertical displacements of node C; all the numerical solutions coincide, even considering pore pressure at node B (see Figure 5d) except for the lowest peak value. Figure 5c describes the quasi-elliptical (or “eight type”) motion of node A due to the Rayleigh wave generated by the impulsive load. As confirmed by Markert et al. [39], the *uvp–formulation* with a Backward Euler scheme provides stiff results due to the fact that this type of scheme possesses a strong artificial damping (see Jansen et al. [40]), while the *up–formulation* overestimates the displacements field due to the assumption of zero relative acceleration between the two phases. The results of the semi-explicit scheme are closer to the solution of TR-BDF2 scheme and the one with quadratic interpolation appears to be the best.

Figure 6 depicts the time sequence of displacements contour and deformed mesh for the semi-explicit scheme with quadratic finite elements. The slow pressure wave (P, generating radial compression) and the shear wave (S, with shear type deformation) propagating in the soil domain are accompanied by the Rayleigh wave (R–wave) moving at the surface of the medium.

### 4.2. Dynamic Poroelastic Responses of Isotropic Porous Media

3D analyses are needed to evaluate the seismic waves in the soil motion and thus to evaluate the predictive capabilities of the F.E. code. For comparison, we first simulate the wave propagation in a fully saturated, homogeneous, and isotropic poroelastic material and analyze the results. Only the semi-explicit-implicit procedure is used due to the robustness of this scheme to solve dynamic problems with a suitable time step size. Furthermore, the splitting procedure provides for a faster solver and requires less computational effort to solve the system of equations.

A solid square prism of soil (dotted lines of Figure 7) subjected to an impulsive pressure load applied on an area of 1 m2 on the top surface has been considered. Taking advantage of the symmetry of the problem, one quarter of the prism has been modeled only; bottom and lateral surfaces are assumed impermeable, frictionless and restrained along the normal direction, whereas the top surface permeable. The FE model is composed by 8820, 21×21×20 3D, linear and equal order elements, with material parameters listed in Table 2. The external vertical impulsive load is the same as in the 2D benchmark case (Figure 4b), with equal duration; a time step size $\Delta t=10^{-3}\;s$ is used in order to respect the CFL condition.

Three different soil compressibilities: $K_{s}=\infty$, $K_{s}=5.2\times 10^{9}\;\mathrm{Pa}$ and $K_{s}=5.2\times 10^{7}\;\mathrm{Pa}$, corresponding to three different Biot’s coefficients (Equation (Equation 13)): B=1.0, B=0.998 and B=0.846, have been taken into account. No appreciable variations in terms of soil displacement and velocities at node A (Figure 8a,b) are visible, with slight differences in terms of fluid velocity (Figure 8c). Far from the impulsive load, the smallest solid compressibility causes wider movements especially along the propagation direction of the impulsive load. When considering an isotropic material, the Rayleigh waves along X and Y presents the same shape and magnitude (see Figure 8d). In Figure 8e, the pore-fluid pressure evolution at nodes E and G is plotted: in case of incompressibility, the peak pressure occurs simultaneously with the external load and then dissipates during the analysis. In the case of larger compressibility, a delay in the peak appears.

By considering the highest compressibility ($K_{s}=5.2\times 10^{7}\;\mathrm{Pa}$), the deformation states of Figure 9 show once again the triggering and evolution of shear and Rayleigh waves, already visible through 2D analysis.

### 4.3. Dynamic Poroelastic Responses with Transversely Isotropic Porous Media

The behavior of an anisotropic and fully saturated porous material replicated by an elastic transversely isotropic constitutive model for both solid material (intrinsic) and porous matrix (structural) is here considered (see Figure 10a), with material parameters shown in Table 3.

#### 4.3.1. Effect of Different Rotation in a Transversely Isotropic Symmetry Axis of Soil Material

Due to structural anisotropy, half of the full 3D domain (dashed blue line) has been taken into account. We assume to rotate by α the *y*-axis of the isotropy plane (solid phase and porous material) with respect to the *y*-axis of the global reference system (see Figure 10). A series of analyses have been performed changing from $\alpha =0{}^{\circ }$ to 90${}^{\circ }$, as shown in Figure 10b. From Equation (Equation 11), the rotation of the isotropic plane activates the coupling components of the elastic (or compliance) tensor as schematically represented in Figure 11, and then it will conduct to different soil responses. The coaxiality assumption between the structural and intrinsic tensor leads to obtaining, as usual, a constant Biot’s coefficient tensor a=BI=0.998I, so the fluid pressure interacts only with the volumetric stresses. The permeability tensor is also anisotropic and coaxial with the direction of material anisotropy; the permeability constants are shown in Table 3, the lowest permeability assumed along the *z*-direction.

A summary of the main results considering three different rotations is shown in Figure 12. By increasing α, the soil stiffness along with *z* decreases, and it increases along with *y*. This leads to an increase in peak values of the vertical displacement at node A (Figure 12a) and in the solid velocity field (Figure 12b), together with a decrease in the peak values of fluid velocity at node A (Figure 12c). Evidently, by considering no rotation of the isotropic plane, α=0, the motion of nodes B and D is the same (black continuous line; Figure 12d,e), while increasing α leads to different motions. This allows for obtaining two different Rayleigh waves along *x* and *y*, being Rayleigh waves linear combinations of P- and S- waves at the surface. In Figure 12f, the surface motion at node C is plotted: it can be observed that, by increasing α, a horizontal motion orthogonal to the wave propagation appears, so along direction *x*=*y*, representing a Love wave (horizontal shifting) that consequently is coupled with the Rayleigh one.

In Figure 13, the pore pressure evolution is reported for two different pairs of nodes (close and far from the impulsive source respectively, see Figure 7) belonging to orthogonal directions: a comparison with the isotropic case leads to observing that the behavior is now strongly different (and no longer superimposed, see Figure 8e), with aligned peak values evidencing a high speed pressure velocity.

By analyzing soil deformation (Figure 14) where $\alpha =90{}^{\circ }$, the different shear waves propagate with different speed and magnitude within the soil medium; hence, the shear wave on the plane YZ reaches the boundary before the one on the XZ plane.

An alternative way to appreciate the shear wave splitting is plotting the effective shear stresses evolution along two planar orthogonal directions $a=\{\forall \left(x,y,z\right)\in R^{3}|x\in R,y=10.5m,z=9.5m\}$ and $b=\{\forall \left(x,y,z\right)\in R^{3}|y\in R,x=10.5m,z=9.5m\}$, Figure 15, considering two different rotations of the material axis. For $\alpha =0.0{}^{\circ }$ (continuous solid line), the propagation of effective shear stresses along *a* and *b* is the same, see, in fact, solid lines of Figure 15a,b and those of Figure 15c,d; whereas, considering $\alpha =45{}^{\circ }$ (marked dashed line), a different behavior is visible. By varying the angle, the shear stress $\tau_{xy}$ speed decreases along *a* but increases along *b*. In the case of mutual stresses $\tau_{xz}$ and $\tau_{yz}$, both speeds increase in different ways.

Particularly, Figure 16 summarizes the reported behavior, i.e., normalized peak stresses and their relative velocity show and confirm a measure of shear wave splitting.

#### 4.3.2. Effect of Biot’s Effective Stress Coefficient Tensor on Wave Propagation

By adopting the same geometric model as before, a series of analyses have been additionally performed by fixing the material parameters of the porous material and rotating the plane of isotropy of the solid phase (intrinsic anisotropy), from $\alpha =0{}^{\circ }$ to 90${}^{\circ }$ as shown in Figure 10c. This assumption leads to obtaining a *full* Biot’s coefficient tensor:(44)$$A\left(\alpha \right)=I-C\;\cdot \;\left[M^{-T}\left(\alpha \right)S_{S}\left(0{}^{\circ }\right)M^{-1}\left(\alpha \right)\right]=\left[\begin{array}{ccc} a_{11}\; & 0\; & 0\\ 0\; & a_{22}\; & a_{23}\\ 0\; & a_{23}\; & a_{33} \end{array}\right]$$
with M(α) the fourth-order transformation tensor and $S_{S}\left(0{}^{\circ }\right)$ the fourth-order compliance constitutive tensor of the solid phase only. In this case, two non zero extra-terms are obtained and, together with the values on the diagonal, the fluid part interacts with the deviatoric stress of the soil. For the initial case of $\alpha =0{}^{\circ }$, the material parameters of the intrinsic transversely isotropic model are listed in Table 4. The anisotropic permeability remains the same as in the previous analyses, and it is coaxial with the symmetry axis of the structural anisotropy.

By varying the Biot’s tensorial coefficients through α, no differences are appreciable in terms of vertical displacements (Figure 17a) and vertical fluid velocity (Figure 17b) at node A, while two different plane motions at nodes B and D are clearly visible. These latter graphs allow us to catch the Rayleigh waves spreading horizontally and to estimate the splitting of the shear waves. In case of high compressibility, different fluid pressure velocities are shown (see Figure 17e,f), the values being no longer aligned as previously evidenced.

By considering a rotation of 90${}^{\circ }$ (this one only for sake of brevity), Figure 18, the splitting of the shear wave is again visible, although less evident than in the previous section. The deformed mesh and the Euclidean norm of the displacements in different time steps of the analysis are plotted considering the solid phase rotated by 90${}^{\circ }$ with respect to the symmetry axis of transversely isotropic model of the mixture (structural anisotropy). A higher solid stiffness phase along Y leads to a decrease in the effective stress along the same direction, provoking a slower shear waves along Y and faster along X.

The results in terms of effective shear stresses (Figure 19a) confirm the behavior already appreciable when rotating the structural anisotropy; when choosing e.g., the second invariant of the Biot’s coefficient tensor as reference (Figure 19b), it can be noticed that higher angles correspond to lower values, with an anomalous splitting mechanism.

## 5. Discussion

The results indicate that the numerical model is able to replicate the following physical phenomena, i.e.,:(i)the p-waves produce polarized vibrations along the direction of propagation (particles move along the wave’s direction of propagation) and subsequent compression and extension deformations along the same direction: they are visible along the vertical direction under the impulsive load (Figure 6 and Figure 9), even considering the anisotropic models (in this case they are coupled with the shear contribution, Figure 14 and Figure 18);(ii)p-waves are faster than s-waves: in all the models in fact the domain borders are reached in different times;(iii)s-waves generate polarized vibrations on a plane containing the direction of propagation and shear deformation (Figure 6, Figure 9, Figure 14 and Figure 18);(iv)the s-wave decouples into a wave polarized on the horizontal plane and into another one on the vertical plane: visible in the curves of effective shear stresses, Figure 15;

Particularly, for s-waves, it has been evidenced that, when propagating vertically, they are always polarized on a vertical plane, whereas, in case of horizontal propagation, one component of the wave belongs to an horizontal plane, the other on a vertical plane. This particularly happens when isotropy or transversely isotropy with vertical isotropy axis is assumed; otherwise, they show inclined polarizations.

With regard to the surface waves, the Rayleigh waves have been properly reproduced: such waves propagate according to cylindrical wavefronts; the resulting motion on the vertical plane is retrograde elliptical (Figure 12d,e and Figure 17c,d) (compare, e.g., with Yang and Li [44]). If considering Love waves (Figure 12f, the generated horizontal vibrations clearly appear polarized along a direction orthogonal to the propagation one (shear deformations).

Even the geometric attenuation of the waves seems to be properly reconstructed: their energetic content being reduced at a far distance from the source, the amplitude of the medium displacement correspondingly decreases (geometric damping): this is evidenced in the effective shear stress curves themselves, Figure 15 and Figure 19a,b. More importantly, when numerically reproducing an anisotropic soil domain, the physical phenomenon of waves splitting appears to be reproduced as well (Figure 16 and Figure 19c,d) with generation of waves with different intensity and speed.

## 6. Conclusions

The propagation of waves in soils, developing from a point source of a dynamic load, have been analyzed with attention focused on polarization and shear wave splitting due to anisotropy of the permeability tensor, anisotropy of the solid skeleton, as well as to a novel Biot’s tensor. The mathematical-numerical model adopts a u–v–p formulation enhanced by the introduction of Taylor–Hood mixed finite elements and comparisons with different integration strategies have revealed to prefer a semi-explicit/implicit scheme with equal order interpolation due to its satisfied stability requirements. The numerical implementation of both an anisotropic permeability tensor together with a Biot’s tensor has allowed for boosting the contribution of anisotropy when reproducing waves polarization and splitting: the conducted analyses have correctly reproduced, to recall a few details, polarized vibrations along the direction of propagation produced by P-waves, polarized vibrations on a plane containing the direction of propagation and shear deformation generated by S-waves, the generation of surface waves, and, more importantly, waves splitting due to intrinsic or structural anisotropy, with enhanced effects when considering the coupling between the volumetric fluid pressure and shear stresses. 

## Figures and Tables

**Figure 1 materials-13-04988-f001:**
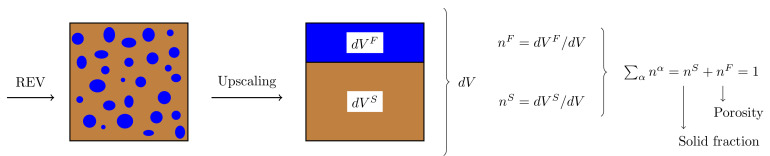
REV for fully saturated porous material.

**Figure 2 materials-13-04988-f002:**
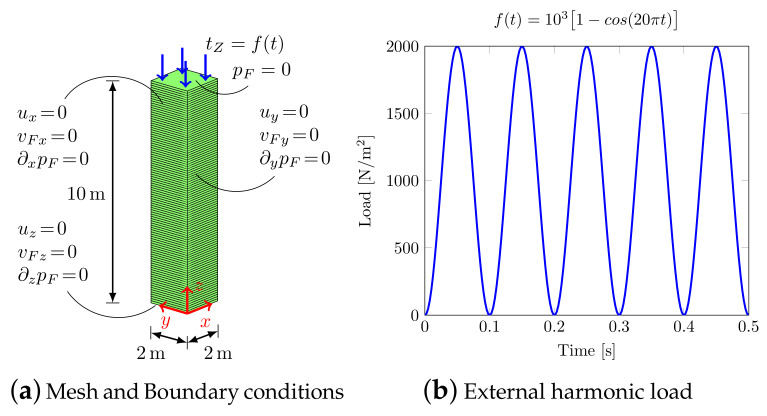
1D model (benchmark (1)).

**Figure 3 materials-13-04988-f003:**
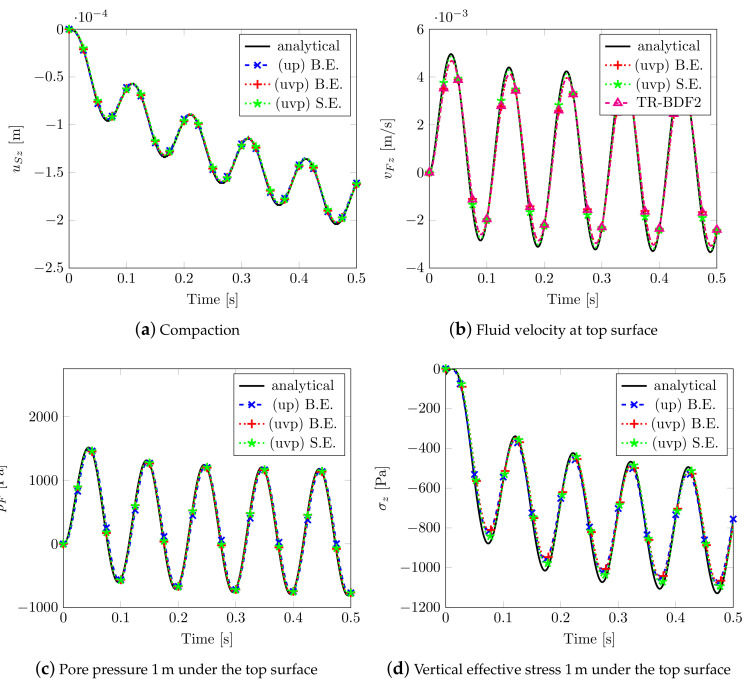
Solutions for Benchmark (1).

**Figure 4 materials-13-04988-f004:**
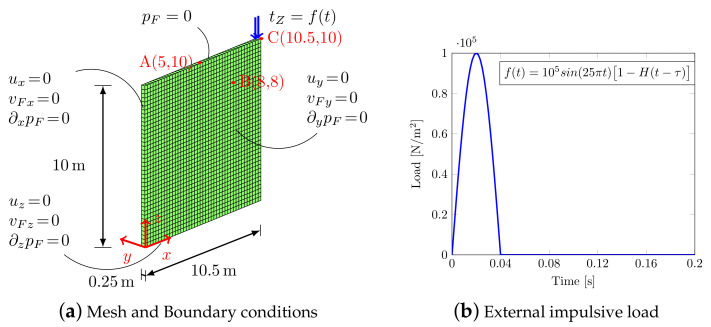
2D model (Benchmark (2)).

**Figure 5 materials-13-04988-f005:**
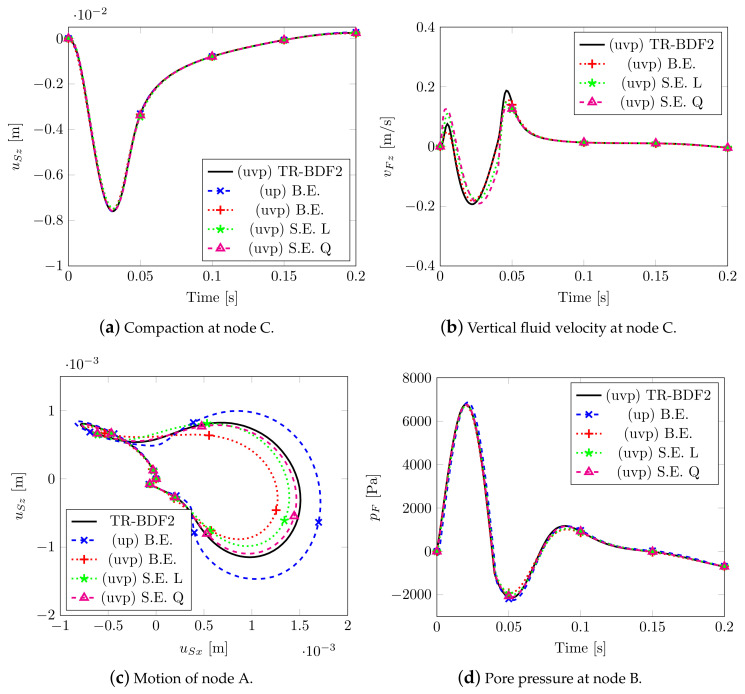
Solutions 2D wave propagation.

**Figure 6 materials-13-04988-f006:**
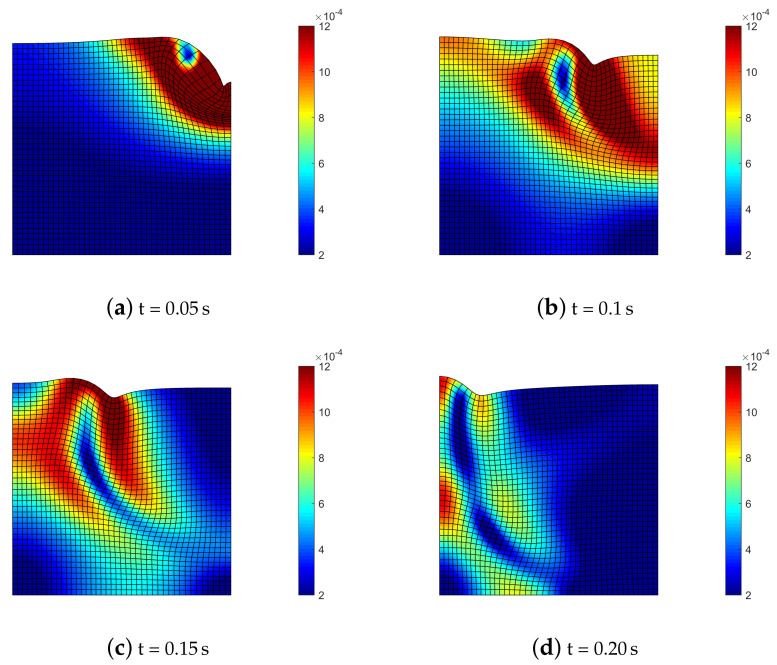
Deformed mesh (amplified by scale factor 500) and contours of norm of soil displacements $∥u_{S}∥=\sqrt{u_{Sx}^{2}+u_{Sz}^{2}}$ for benchmark (2).

**Figure 7 materials-13-04988-f007:**
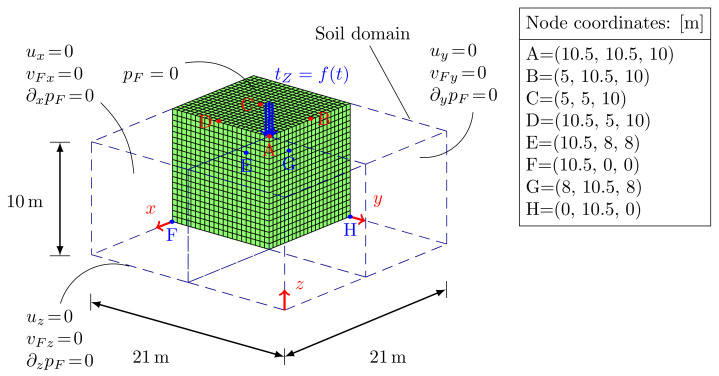
Three-dimensional soil model.

**Figure 8 materials-13-04988-f008:**
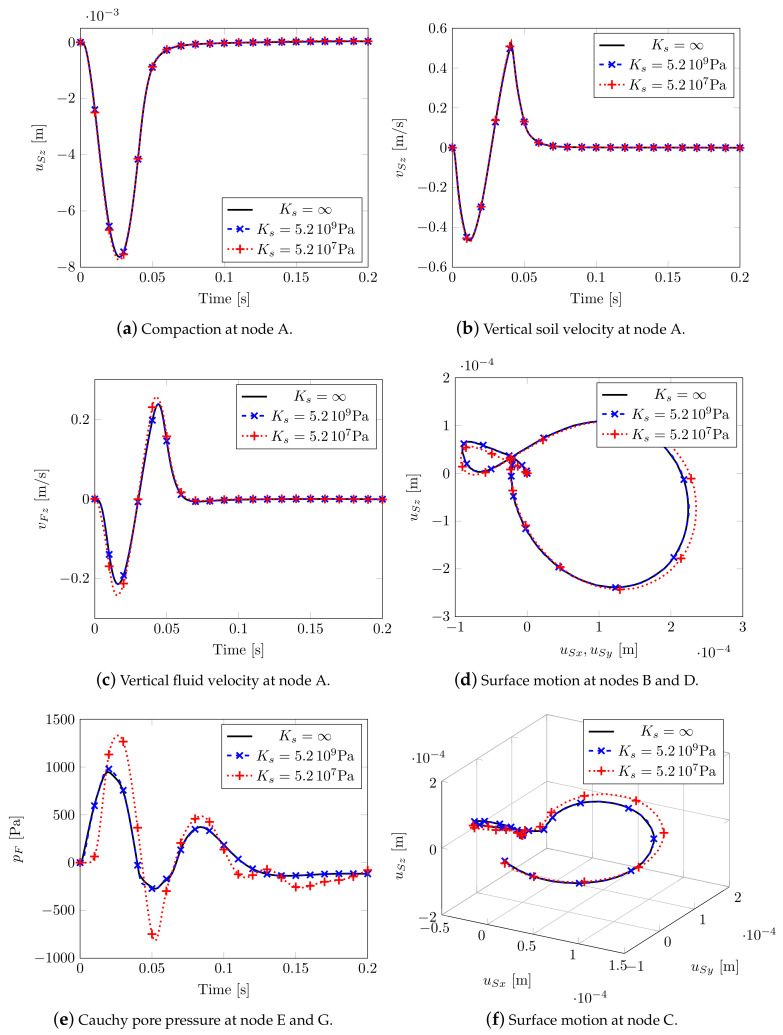
3D isotropic soil model.

**Figure 9 materials-13-04988-f009:**
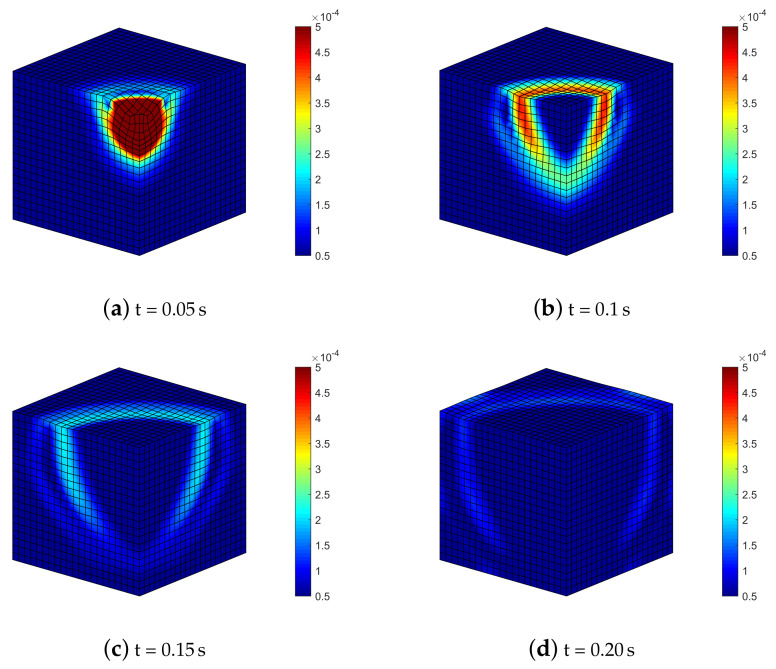
Isotropic soil model with $K_{s}=5.210^{7}Pa$: deformed meshes (amplified by scale factor 500) and contours of norm of soil displacements vector $∥u_{S}∥=\sqrt{u_{Sx}^{2}+u_{Sy}^{2}+u_{Sz}^{2}}$.

**Figure 10 materials-13-04988-f010:**
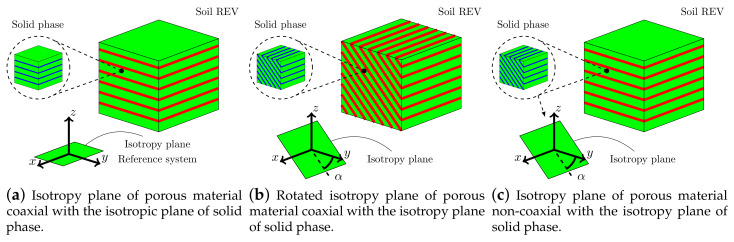
Schematic diagram of transversely isotropic soil models.

**Figure 11 materials-13-04988-f011:**
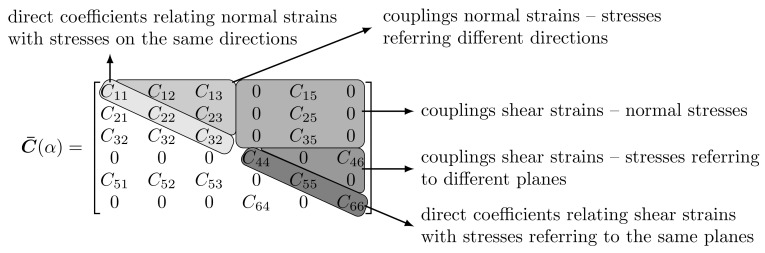
Subdivision of the elastic matrix.

**Figure 12 materials-13-04988-f012:**
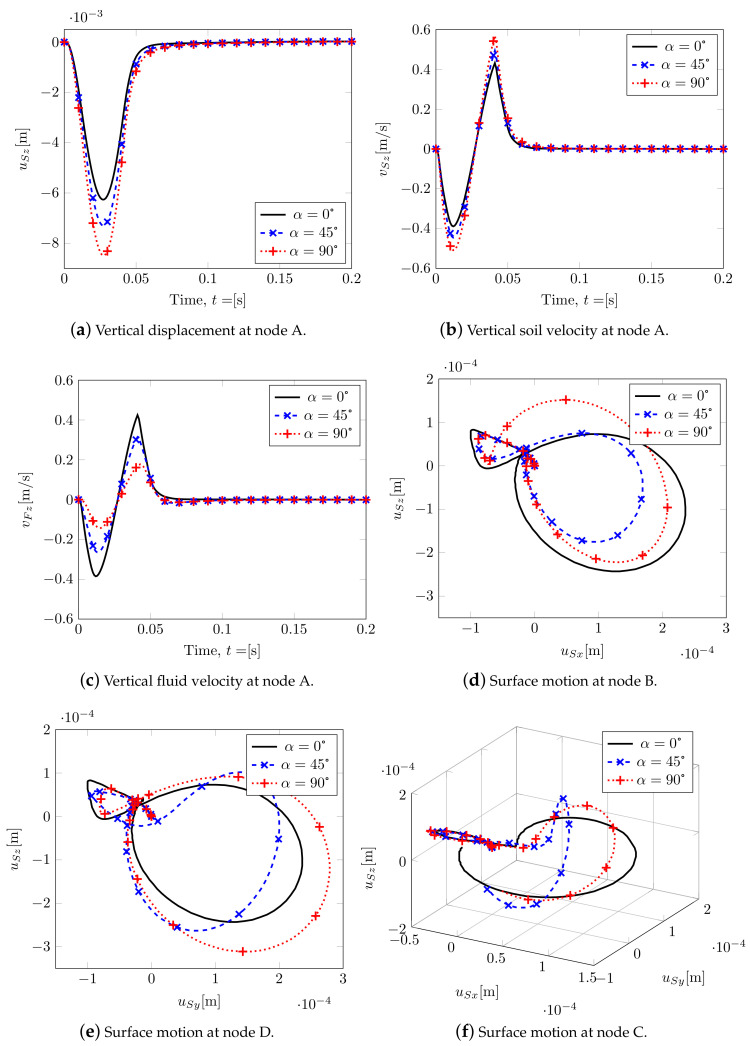
Transversely isotropic soil model. Displacements and velocities.

**Figure 13 materials-13-04988-f013:**
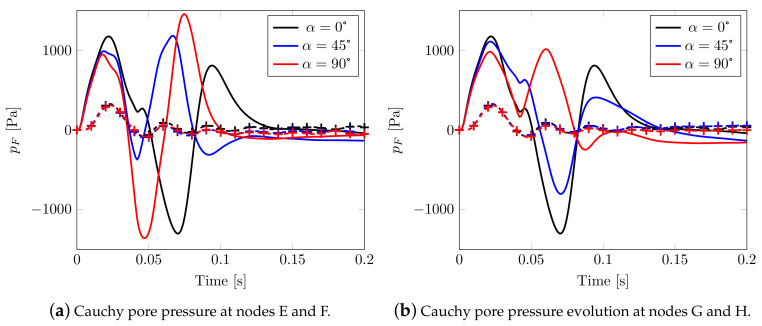
Cauchy pore pressure evolution in time: solid line for nodes E and G, marked line for nodes F and H.

**Figure 14 materials-13-04988-f014:**
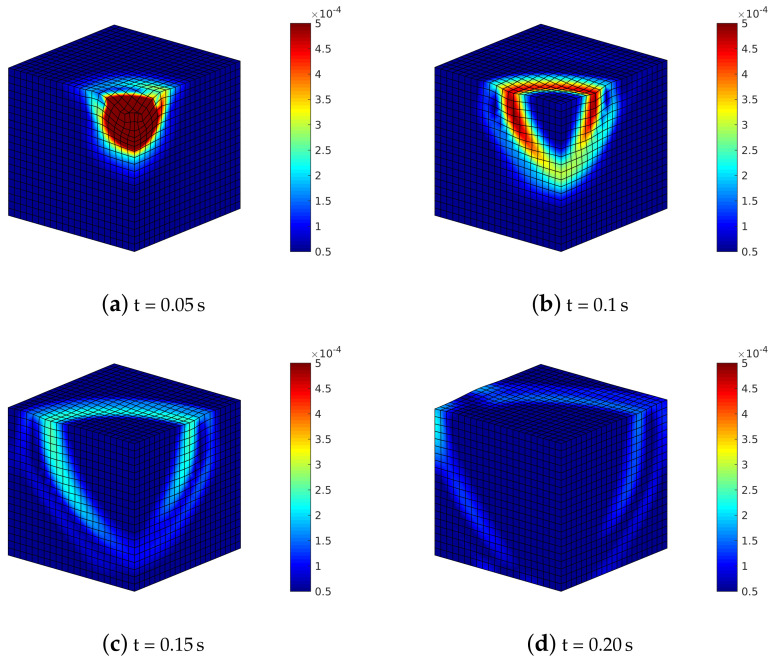
Transversely isotropic soil model with $\alpha =90{}^{\circ }$: deformed meshes (amplified by scale factor 500) and contours of norm of soil displacements vector $∥u_{S}∥=\sqrt{u_{Sx}^{2}+u_{Sy}^{2}+u_{Sz}^{2}}$.

**Figure 15 materials-13-04988-f015:**
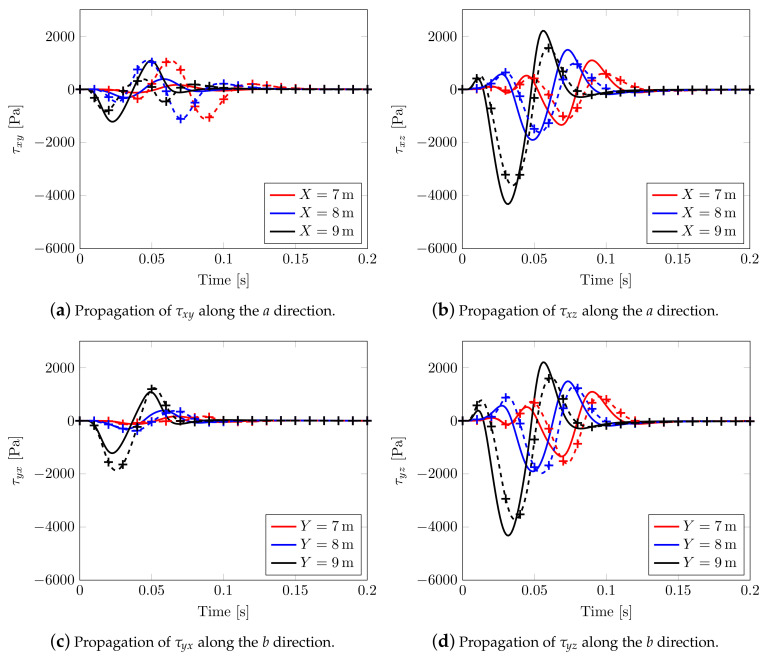
Transversely isotropic soil models. Effective Cauchy stress component calculated along the horizontal directions: solid line for model with $\alpha =0{}^{\circ }$, marked line for model with $\alpha =45{}^{\circ }$.

**Figure 16 materials-13-04988-f016:**
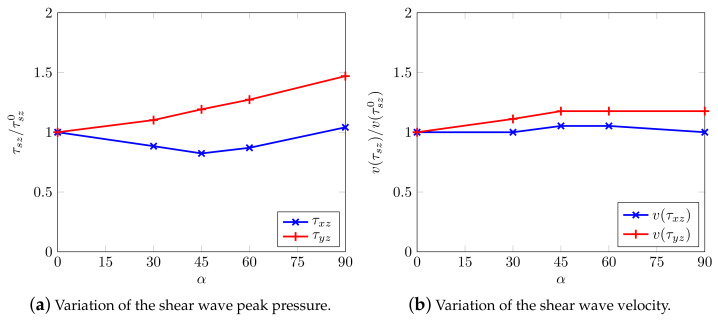
Shear wave splitting along the horizontal axis.

**Figure 17 materials-13-04988-f017:**
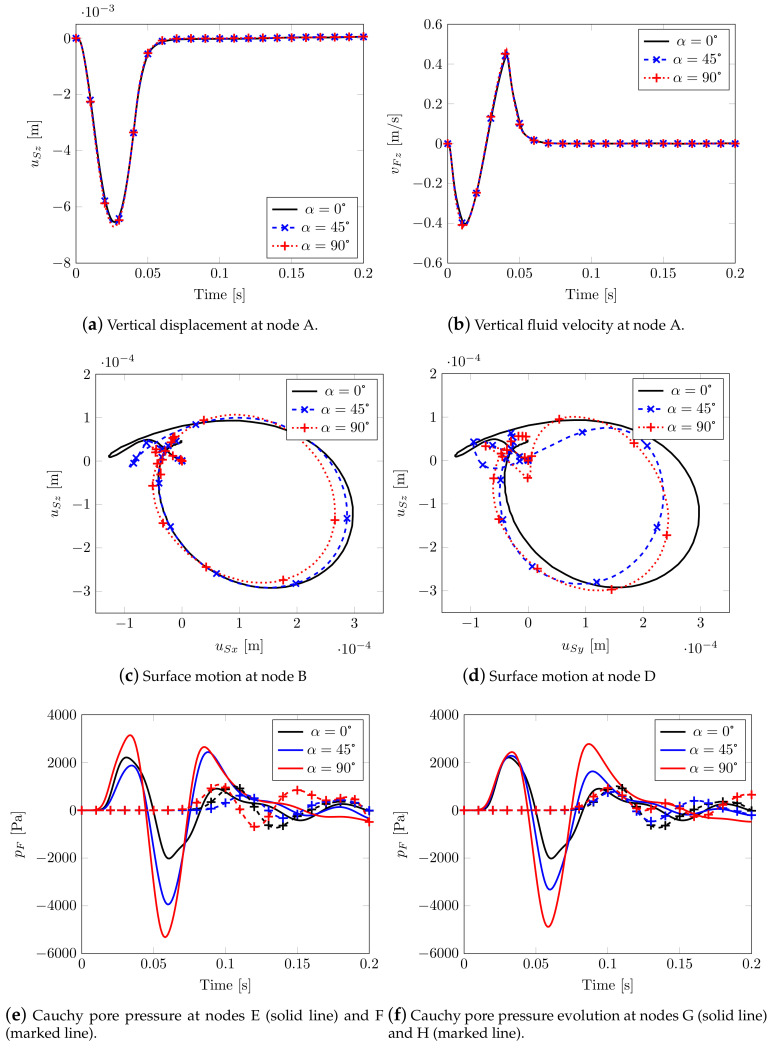
Effect of Biot’s coefficient tensor: main variables.

**Figure 18 materials-13-04988-f018:**
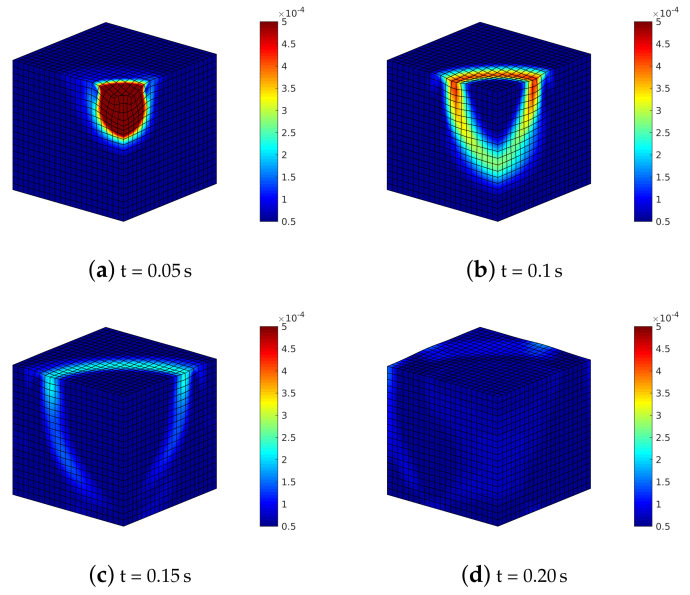
Transversely isotropic soil model with constitutive model of solid phase rotated by $\alpha =90{}^{\circ }$: deformed meshes (amplified by scale factor 500) and contours of norm of soil displacements vector $∥u_{S}∥=\sqrt{u_{Sx}^{2}+u_{Sy}^{2}+u_{Sz}^{2}}$.

**Figure 19 materials-13-04988-f019:**
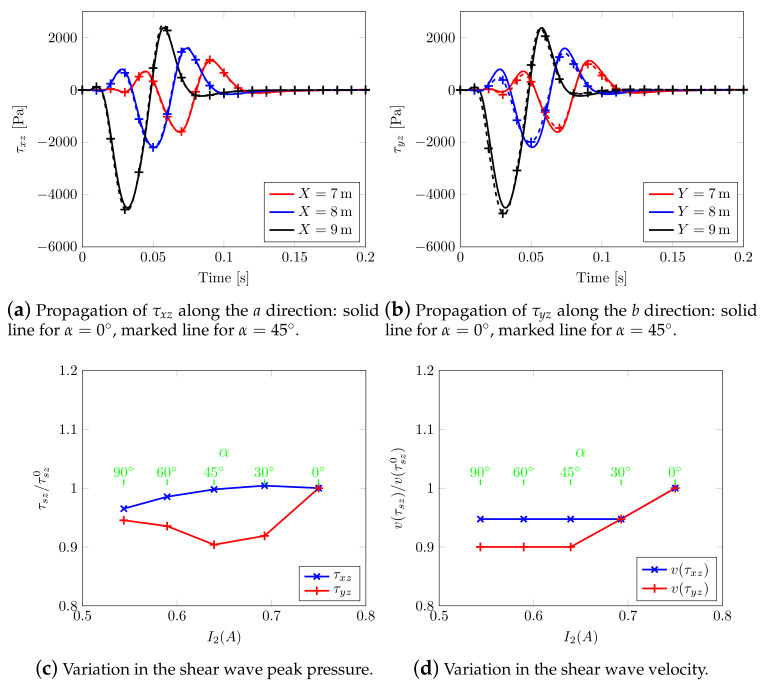
Effect of Biot’s coefficient tensor: effective shear stress component calculated along the horizontal directions.

**Table 1 materials-13-04988-t001:** Material parameters for benchmarks (1) and (2).

Parameter	Values	S.I. unit
*E*	14.52 × 10${}^{6}$	Pa
ν	0.30	
$n_{0}^{F}$	0.33	
$k^{F}$	10${}^{-2}$	m/s
$\rho_{S}$	2000	kg/m${}^{3}$
$\rho_{F}$	1000	kg/m${}^{3}$

**Table 2 materials-13-04988-t002:** Material parameters for the 3D model.

Parameter	Values	S.I. Unit
*E*	12.0 × 10${}^{6}$	Pa
ν	0.25	
$n_{0}^{F}$	0.33	
$k^{F}$	10.0${}^{-2}$	m/s
$\rho_{S}$	2000.0	kg/m${}^{3}$
$\rho_{F}$	1000.0	kg/m${}^{3}$
$K_{F}$	5.2 × 10${}^{9}$	Pa

**Table 3 materials-13-04988-t003:** Transversely isotropic soil: material parameters for the 3D model.

Parameter	Values	S.I. Unit
$E_{x},E_{y}$	9 × 10${}^{6}$	Pa
$E_{z}$	15 × 10${}^{6}$	Pa
$\nu_{xy},\nu_{yx}$	0.25	
$\nu_{yz},\nu_{xz}$	0.21	
$\nu_{zx},\nu_{zy}$	0.35	
$G_{xy}=\frac{E_{x}}{2(1+\nu_{xy})}$	3.6 × 10${}^{6}$	Pa
$G_{23},G_{31}$	6.0 × 10${}^{6}$	Pa
$n_{0}^{F}$	0.33	
$k_{x}^{F},k_{y}^{F}$	10${}^{-2}$	m/s
$k_{z}^{F}$	10${}^{-4}$	m/s
$\rho_{S}$	2000	kg/m${}^{3}$
$\rho_{F}$	1000	kg/m${}^{3}$
$K_{m}$	7.14 × 10${}^{6}$	Pa
$K_{S}$	3.57 × 10${}^{9}$	Pa
$K_{F}$	2.2 × 10${}^{9}$	Pa

**Table 4 materials-13-04988-t004:** Intrinsic transversely isotropic model: material parameters.

Parameter	Values	S.I. Unit
$E_{x},E_{y}$	1.8 × 10${}^{7}$	Pa
$E_{z}$	3.0 × 10${}^{7}$	Pa
$\nu_{xy},\nu_{yx}$	0.25	
$\nu_{yz},\nu_{xz}$	0.21	
$\nu_{zx},\nu_{zy}$	0.35	
$G_{xy}$	7.2 × 10${}^{6}$	Pa
$G_{yz},G_{zx}$	1.2 × 10${}^{7}$	Pa

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
