# Peer review of "Shear Wave Splitting and Polarization in Anisotropic Fluid-Infiltrating Porous Media: A Numerical Study"

_materials, 2020, doi:10.3390/ma13214988_

Round 1

Reviewer 1 Report

The manuscript addressed an interesting topic on shear wave splitting and polarization in anisotropic fluid-infiltrating porous media. The study has been mainly focused on the numerical formulation especially the numerical time integration strategies. Overall, the topic should be of interest to many readers, and the presentation is clear and well organized. The manuscript could be of high quality if the authors would like to spend more time on explaining the underlying physics associated with the anisotropic fluid-infiltrating porous media. Some comments are provided for consideration.

(1) Starting from Line 27, the authors mentioned that shear-wave splitting contains information about the orientation of crack distribution. Please provide some reference for the benefit of readers. For instance, Grechka, Kachanov, and many other researchers have tried to address this issue and to recover the crack distribution information from the measurement of anisotropic elastic waves in the past decades.

(2) There is inconsistency in terms of symbols throughout the manuscript. For instance, the symbol of the effective stress in Eq. 12 is not the same as elsewhere in the manuscript. Those are small issues but could be easily improved. Indeed, it is better if the authors could provide a list of symbols to help keep the readers’ attention.

(3) It is not entirely clear for the reviewer how to interpret the Biot’s effective stress tensor presented in Eq. 12. To the reviewer’s understanding, having the off-diagonal entries in the Biot’s effective stress tensor suggests that the pore pressure has an implication on the deviatoric stress. This is not intuitive and may also confuse some readers. It is better that the authors could explain the concept in more details. In addition, since the Biot’s effective stress coefficient and permeability/hydraulic conductivity constants are extended to tensors, it is worthwhile to mention the other constants that have been heavily used in the Biot’s theory. Presumably, the specific storage coefficient or Biot modulus should be also extended to tensors.

(4) Similar to (3), does that make sense to further generalize Eq. 14 such that the directionality of “porosity” could be incorporated?

(5) If possible, when deriving the stability criterion and critical time-step (Eq. 43), avoid using the Lame constants, which are only a simple for isotropic medium, but use more general representation such as the eigenvalues of the stiffness/compliance tensor because the anisotropic medium is the concern of this manuscript.

(6) At the beginning of Section 4 (from Line 205), the authors mentioned that they examined the case of the induced anisotropy related to volumetric-deviatoric coupling effects. Maybe the reviewer misunderstood it, but this part was not found in the following sections.

(7) In 2D and 3D numerical examples, the authors presented only the numerical solutions but not comparisons with an analytical solution or experimental data. The reviewer understood that it may not be easy to find an analytical solution to the 2D or 3D cases, but it is hard to demonstrate the validity of the numerical schemes without such comparisons. At least, the authors could compare the wave speeds.

(8) The author mentioned that the solution of TR-BDF2 scheme provided the best results of the Rayleigh wave presented in Fig. 5(c). How do the authors prove one of the schemes is the best without comparing with the “ground truth”? In particular, the authors mentioned the uvp-formulation with Backward Euler scheme provides stiff results due to the fact that it posses strong artificial damping. Could the authors elaborate it a little bit more?

(9) Section 4.2 is supposed to explore the poroelastic responses of isotropic media. Overall, the reviewer liked it but it could be better if the authors could address more poroelasticity effects. Did the authors observe the change of elastic responses such as wave speed over time? It is supposed that the porous medium under rapid loading could exhibit undrained responses in a short time frame and drained responses in the long-time frame. It is interesting to see if this could be reflected by the change of the calculated wave speeds.

(10) Section 4.3.2 is interesting. Again, as mentioned in Comment #3, it is not intuitive that the porous material and the solid phase is not coaxial. How does such non-coaxiality between so-called structural anisotropy and intrinsic anisotropy associate with the microstructure of the material? Numerically, it is of course that some interesting results could emerge, but it is important to relate them to the physics and material.  

Author Response

Dear Reviewer,

thanks for your comments, the answers are listed below:

  1. Some references dealing with the relation between shear-wave splitting and crack distribution have been added. Please see lines 45-48.
  2. Symbols have been rechecked. Please note that the effective stress is throughout indicated with subscript (S) and superscript (E) apart from Figs. 3(b), 15, 16 and 19 just for giving a more schematic representation.
  3. Yes, when Biot’s tensor is full, pore pressure can produce deviatoric stresses. A few lines have been added after Eq.(12). Apart from Biot’s and permeability tensors (the latter already being a tensor even in isotropic conditions), the compressibility modulus remains a scalar but its expression changes according to Eq. (27).
  4. The “directionality” of porosity is given indirectly via the permeability and the constitutive tensors but, at least according to our approach, its nature can be considered overall scalar being the void percentage a scalar itself.
  5. Yes, thank you, it was a typos, now Eq. (43) has been corrected considering the anisotropic case.
  6. The case of the induced anisotropy related to the volumetric-deviatoric coupling has been studied in section 4.3.1: the rotation of the constitutive tensor of the porous material activates the extra-diagonal coupling term (see Fig. 11) and leads to an anisotropic behavior along directions X and Y.
  7. Both 1D and 2D cases have been compared with literature results; particularly, the 2D numerical results fully agree with those by of Market [2010]. Being exactly superimposed, we decided not to plot them, both for brevity reasons and for concentrating on the different solution algorithms. A statement has been added in § 4.2. Differently, the 3D model is treated as (after having validated it) a predictive model; please see the Section 5 “Discussion” which has been added on purpose.
  8. As written in row 260 and below the TR-BDF2 scheme satisfies all the stability requirements and it is second order accurate [Markert 2010], so it has been taken as reference solution for the 2D problem (being unavailable experimental results). So, the semi explicit scheme with quadratic interpolation appears to be the best just being its results closer to the TR-BDF2 one. Additionally, the numerical damping coming from a standard Backward Euler scheme has been discussed elsewhere (see the new reference added in rows 272-273).
  9. Yes, it is possible to obtain such changes (some are already visible, please see the Discussion Section), but from one side the domain is not wide enough to ensure no reflecting waves, from the other side the assumed permeability values are too high for appreciating drained-undrained responses. Additionally, an experimental dataset should be necessary to effectively prove the obtained numerical values. At present our objective was different, i.e. to prove that the Finite Element model can appropriately reconstruct physical phenomena, particularly wave splitting with generation of both volumetric and superficial waves.
  10. Just to give an example: the intrinsic anisotropy comes e.g. from a non-porous material behaving anisotropic from the mechanical point of view; if now we add pores, voids or faults, the whole porous medium acquires an anisotropy which is generally different (so, non-coaxial) from the one of the solid phase and this stands for structural anisotropy

Reviewer 2 Report

The topic is interesting and important to “Shear wave splitting and polarization in anisotropic fluid-infiltrating porous media: a numerical study”. The purpose of this manuscript is important and numerical results are interesting. However, unfortunately, the manuscript is difficult to understand due to insufficient presentation. The manuscript cannot be recommended for publication in its current state. I have carefully reviewed the paper. I think this paper has not an enough quality to be accepted by “materials” in current state.

Here below please find a series of comments that the authors may want to consider to improve the clarity of their manuscript

  • Abstract: The abstract is weak and need to be enhanced. Results are not outlined in the abstract. Authors must include briefly their findings.

  • Main purpose of the contribution?

What I am missing is a clear message – if authors have 1 minute to tell someone why this work is exciting and why they should read it – what would you say. Right now it is not clear. For that it is actually really important to write the abstract – What is it that authors think is the main advanced with this work?

  • We use normally the introduction to bring the reader up to speed what is known in the subject area and where the gaps are. Authors need to work on the introduction as this does not come across: mathematical model and numerical implementation, shear wave splitting and polarization in anisotropic media mechanics behind shear wave splitting, and where the gaps are.
  • The authors should explain clearly about the shear wave splitting and polarization in porous media.

  • All the results and discussion are rather general, should be explained clearly and scientifically.

  • The results/discussion needs a lot of work in my opinion, the first part has no references, authors need to compare to literature much more. I think here it would be important to have a section that assesses the mechanism of shear wave splitting, knowledge out of them.

  •  The authors should also review the conclusions thoroughly.

  • Paper written language sometimes is a bit confusing, watch your punctuation, keep to the scientific writing style

Minor revision is necessary with more scientific expiation.

Author Response

Dear Reviewer,

thanks for your comments; our modifications are briefly summarized below (please consider the revised version of the paper with new parts in red):

  • Abstract has been rewritten.
  • The main purposes have been added in the Introduction, together with a clearer explanation of wave splitting, polarization and comments on the numerical results (see also the new “Discussion” Section).
  • Conclusions have been extended.
  • Language has been revised.

Reviewer 3 Report

The manuscript proposed by De Marchi et al. entitled "Shear wave splitting and polarization in anisotropic fluid-infiltrating porous media: a numerical study" presents an interesting analysis on wave propagation and shear wave splitting in fully saturated porous media that exhibits a variety of anisotropy. Besides, some "recent" numerical approaches have been conducted.

In my view, before its publication, some aspects should be addressed.

-Abstract section is very short and should be extended.

-English editing should be carried out, regarding comma usage.

-More recent references should be added, since just one is after 2017, and it is a self-cite. For instance, related to shear wave,

https://doi.org/10.3390/ijgi8070309

-Conclusions section should be extended.

-References section should be fully rewritten to fulfill the journal style. I recommend using a reference manager editor, such as Mendeley, Zotero, or so forth.

Author Response

Dear Reviewer,

thanks for your comments; the main changes are reported below, please consider the revised version of the paper:

  1. Abstract has been rewritten.
  2. The use of the comma was checked throughout the article.
  3. More recent references have been added to the lines 22, 23, 67, 234, 398.
  4. The conclusions have been extended.
  5. The bibliography has been rewritten following the journal style.
